# Improvement of the Non-Destructive Testing of Heritage Mural Paintings Using Stimulated Infrared Thermography and Frequency Image Processing

**DOI:** 10.3390/jimaging5090072

**Published:** 2019-08-29

**Authors:** Kamel Mouhoubi, Vincent Detalle, Jean-Marc Vallet, Jean-Luc Bodnar

**Affiliations:** 1ITHEMM, UFR Sciences Exactes et Naturelles, BP 1039, 51687 Reims CEDEX 02, France; 2C2RMF, 14 Quai François Mitterrand, 75001 Paris, France; 3CICRP, 21 rue Guibal, 13003 Marseille, France

**Keywords:** stimulated infrared thermography, work of art, post-processing, Fourier transform, image enhancement, NDT

## Abstract

Within the framework of conservation and assistance for the restoration of cultural property, a method of analysis assistance has been developed to help in the restoration of cultural heritage. Several collaborations have already demonstrated the possibility of defects detection (delamination, salts) in murals paintings using stimulated infrared thermography. One of the difficulties encountered with infrared thermography applied to the analysis of works of art is the remanence of the pictorial layer. This difficulty can sometimes induce detection artifacts and false positives. A method of thermograms post-processing called PPT (pulse phase thermography) is described. The possibilities offered by the PPT in terms of reducing the optical effects associated with the pictorial layer are highlighted first with a simulation, and then through experiments. This approach can significantly improve the study of painted works of art such as wall paintings.

## 1. Introduction

Heritage conservation is an important societal challenge. The uniqueness and value of cultural heritage involves as much effort as possible to implement non-destructive testing techniques for diagnosis purposes. A lot of them are tested, including stimulated infrared thermography [1,2,3,4,5,6,7,8,9,10,11,12,13,14,15]. “Laboratoire de Recherche des Monuments Historiques” (LRMH), “Centre Interdisciplinaire de Conservation et Restauration du Patrimoine” (CICRP) and “Institut de Thermique, Mécanique, Matériaux” (ITheMM) of the Reims University (URCA) are thus collaborating in order to develop this last one. First results showed that stimulated infrared thermography allows detecting detachments within wall paintings [7,8]. However, the sensitivity of this method to the thermal response of pigments can lead to induce artifacts in the detection of defects. This impact can be reduced in our point of view by the way of Pulse Phase Thermography analysis.

This method is based on discrete Fourier transform analysis of the thermographic film obtained. It gives access to two characteristic parameters: The amplitude and the phase of the photothermal signal. The amplitude is potentially sensitive to heterogeneities of energy deposition on it. The phase measures the delay of the photothermal response in relation to the visible excitation as a function of frequency and it is thus potentially less sensitive to surface radiative issues.

Our work has been focused on this last property according to two research axes: We first made a simulation based on finite element modelling of the photothermal experiment which has been developed on an academic sample. We also developed an experimental study on two laboratory samples. The obtained results are presented according to the following four sections:

First, the principle of the implemented photothermal model is developed. In a second step, we present the obtained simulation results and we show that the phase parameter is interesting in terms of rejection of the optical effects induced by the different colors of the pictorial layer. In a third step, the experimental study is described: We present the samples and we describe the experimental system implemented. Finally, we discuss the obtained experimental results and their match to the theoretical conclusions.

## 2. The Photothermal Model Developed for the Experiment

### 2.1. Principle of the Pulse Phase Thermography (PPT)

The principle of stimulated infrared thermography is divided into three steps: First, the sample under analysis is excited by a luminous flux. This energy input causes an increase of the temperature on the surface of the sample. The heat is then diffused into the analyzed sample. The presence of a defect disturbs the thermal diffusion, and leads to a slower cooling of the analyzed surface in the case of a displacement. This last change can be detected by an infrared thermography camera. 

In the case of the PPT in infrared thermography, the excitation signal is of the crenel type. The thermographic film is then subjected to a mathematical post-processing, such as discrete Fourier transform (DFT), that is defined by the following expression:(1)$$F\left(u\right)=\frac{1}{N}\overset{N-1}{\underset{n=0}{\sum }}h\left(x\right)exp\left[\frac{-j2\pi ux}{N}\right]=R\left(u\right)+jI\left(u\right)$$

When T(t) is a measured thermal signal at a pixel, after sampling it gives a succession of measurement points of index *x*, *h*(*x*) represents the temperature for this pixel at the index *x*, *N* is the number of measurement points resulting from the sampling and *u* is the frequency calculated by applying Expression (1). Maldague et al. [16,17,18] introduced this mode of analysis for the thermography. It allows, while being quick (crenel excitation), access to the characteristic parameters provided by a sinusoidal excitation: the amplitude and the phase of the photothermal signal from the real and imaginary components *R*(*u*) and *I*(*u*). They are obtained from Equation (1). Refer to the books [19,20] on the discrete Fourier transform for more information. 

The sequential processing of the thermogram is made pixel by pixel in a fixed time interval. It makes it possible to extract an image of amplitude and phase for each frequency component *u*.

Initially, this variant had the objective of a good noise rejection and consequently a detection of low photothermal signatures. In our case, our interest is about the temporal connotation of the phase. Because it is less sensitive to energy fluctuations, such as heterogeneities of energy deposition. These heterogeneities are induced by different radiative absorptivities coming from the pictorial layer that are present in the raw thermographic film. So, it enables us to detect the defects better. 

### 2.2. Simulation Developed for the Study

The simulation developed for the study is based on solving the heat equation by the finite element method. The process is performed in two steps: first, the calculation engine determines numerically the spatial and temporal distribution of temperature, and second, we collect these data and perform a discrete Fourier transform (DFT) on the surface temperatures of the sample. This finally results in amplitude and phase maps of photothermal signal as a function of frequency. 

The sample we analyzed theoretically (Figure 1) is a block of plaster which has thermophysical properties very similar to those of a mural painting. A Figure 2 shows the meshing of the sample studied, a fine mesh has been chosen.

It is 160 mm by 120 mm in size and 20 mm thick. The retained thermophysical properties are as follows: a thermal conductivity of 0.4 W/mK, a density of 1100 kg/m^3^, a calorific capacity of 830 J/kg K, which gives a thermal diffusivity of 4.38 × 10^−7^ m^2^/s. Six parallelepiped spaces containing air (detachments) were arranged in the volume of this sample. Their dimensions are 20 mm in length and in width and 4 mm in thickness. Their depths vary from 2 mm to 12 mm in 2 mm increments. 

The thermophysical properties taken into account for these defects are those of the air at 20 °C: a thermal conductivity of 0.026 W/mK, a density of 1.17 kg/m^3^, a calorific capacity of 1006 J/kg K or a thermal diffusivity of 2.22 × 10^−5^ m^2^/s.

Finally, two areas were drawn on the surface of the sample in order to simulate the optical effects induced by the pictorial layer: The first area has an absorptivity 1.5 times higher than the second area and covers defects located at 4 mm, 8 mm and 12 mm deep. The second zone covers defects located at 2 mm, 6 mm and 10 mm in depth. The type of excitation signal to the sample is a crenel of 2 seconds duration (Figure 3). The analysis time is 200 s. The acquisition frequency is fixed at 1 Hertz. The power of energy used is 1500 W. The energy distribution was adjusted as follows: the zone of lower absorptivity received a total power of 1000 W while the zone of higher absorptivity received a total power of 1500 W. We considered a no-loss thermal model.

## 3. Results Obtained

In Figure 4, we present the obtained raw thermograms at t = 28 s, t = 91 s, t = 177 s and t = 200 s. They show a higher photothermal heating at the location of the defects.

The intensity of the background is variable and comparable defects in terms of depth do not appear with the same contrast. The Figure 4 show that differences in surface absorptivity can lead to the creation of detection artifacts. We are trying to reduce this phenomenon.

In Figure 5, we then present the results obtained after a PPT-type frequency analysis. It shows the obtained module images, corresponding respectively to frequencies of 5 mHz, 10 mHz, 20 mHz, 30 mHz, 40 mHz and 80 mHz. 

This figure shows that frequency images of amplitude, like raw signals, can detect all 6 defects. It also shows that this detection is biased by the inhomogeneities of energetic deposition due to the pictorial layer; defects at near depths do not appear with the same intensity. As expected, the amplitude analysis does not allow us to reduce the parasitic effects due to the pictorial layer.

Figure 6 shows the phase images of the calculated harmonic response at the same frequencies. The 6 defects located in the analyzed sample are well detected. This phase analysis shows that it has been possible to homogenize the background, even in bands with different optical absorptivity. This result is encouraging, as it seems to reduce the parasitic optical effects induced by the different colors of the pictorial layer. It is now necessary to confirm this experimentally.

## 4. Experimental Study

### 4.1. Samples Description

Based on the results obtained by the theoretical study, an experimental validation of the PPT should be implemented in order to ensure the demonstration. Two samples have been built with different internal defects.

The first one dedicated to our research was built in our laboratory, and is a block of plaster whose thermal properties are close to those of a wall paint. It is covered with a pictorial layer (acrylic paint) composed of 20 rectangular areas of various colors covering the visible spectrum (from purple to red), moreover this choice of pigments allows representing the different cases of figure in terms of absorptivities: The black color has a high absorptivity, the white color is highly diffuse, the silver color is very reflective, and so on. These different colors will have very different behaviors in terms of energy absorption.

The geometric dimensions of this block are 68 cm wide, 50 cm high and 5 cm thick in order to be able to study in one recorded area with a sufficient resolution. 

In order to simulate delaminations, 32 extruded polystyrene inserts covering the entire surface were introduced into this sample (Figure 7 and Figure 8). These are discs with a diameter of 20 mm and a thickness of 2 mm placed at a depth of 5 mm.

The second built sample is a partial copy of the mural painting “Saint Maurille en Evêque” of the Angers Cathedral painted in 1270–1280 (Figure 9). This sample was built and painted according to the painting technics studied before by the research laboratory of historical monuments that discovered one of the first oil paintings of mural painting. It therefore has thermal and optical properties close to the work of art itself.

This copy is shown in Figure 10. It represents the character located to the right of the “Saint Maurille” (white border of Figure 9). We choose this part of the wall painting because it offered different radiative behaviors. The black and dark parts easily absorb the exciter light flux. The lighter parts diffuse it partially. Finally, the gold insert (clover-shaped) behaves like a diffusing mirror. The geometric dimensions of this sample copy are a width and length of 15 cm. Its thickness is equal to 15 mm. To simulate local deterioration of the support stone, 5 extruded polystyrene inserts were introduced into this sample, see Figure 10a. These are 30 mm diameter and 2 mm thick discs. They were placed at variable depths of 2 mm, 3 mm, 4 mm, 5 mm and 6 mm. They were spread as shown in Figure 11.

### 4.2. The Experimental Set-Up Implemented for the Study

Both samples were analyzed using the SAMMTHIR system of the laboratory. This device first includes an excitation by two halogen lamps. It then includes an infrared thermography camera. It is the FLIR SC655 micro-bolometer camera, used for the acquisition of the photothermal signal. It includes an electronic control and data acquisition system, in addition to a data analysis and processing software developed by FLIR ©.

In order to study the first sample, the experimental conditions selected are as follows: the camera is at a distance of about 150 cm from the sample. The 500 W halogen light sources are 45° oriented in the direction of the sample, and they are placed on either side of the camera in order to homogenize the heating flux. The excitation is of the crenel type with a duration time equal to 1 min. The analysis time is equal to 194 s with an acquisition frequency of 1 Hz. Figure 12 shows the first sample during the experiment.

For the second sample, we have chosen experimental conditions close to the previous ones: the infrared thermography camera type SC655 was placed roughly 50 cm from the work of art. The halogen excitation sources were placed on either side of this camera, 50 cm from the sample and inclined about 45° from its surface with an average power of 2 × 500 Watts. The excitation time was equal to 2 min at 1 Hz of frequency (see Figure 13 of the sample under analysis).

### 4.3. Results and Discussion

The first experiments were performed on the plaster block described above. In Figure 14, we first present the raw thermograms obtained at times t = 10 s, t = 60 s and t = 120 s. They clearly show the possibility of detecting, after 120 s, most of the inserts present in the sample studied. At the same time, the beginning of the recording time, they also show very clearly the important influence of the pictorial layer emissivity on the collected photothermal signal.

In Figure 15 and Figure 16, we present the results after a Fourier transform treatment and analysis. Figure 16 shows the obtained module images corresponding respectively to frequencies of 5.15 mHz, 10.30 mHz, and 0.9 Hz. As expected, it shows that these signatures allow a partial detection of the defects, but also that they are very sensitive to inhomogeneities of energy deposition.

Figure 16 shows the obtained phase images at the same frequencies. On one side, we get the possibility of detecting almost all the defects located in the sample studied. On the other side, they show a significant rejection of the optical effects induced by the pictorial layer, which was the desired goal. This method, which is based on the PPT, seems to be really appropriate to reduce the optical effects induced by a pictorial layer.

With this first encouraging experimental result, we were focused on the copy of the “Saint Maurille en Evèque”. In Figure 17, we present the raw thermograms obtained at times t = 1 s, t = 41 s, and t = 108 s. On the one hand, they show a particular photo thermal signature at the place of the 5 defects since it allows their detection. On the other hand, they show the important influence of the pictorial layer on the photothermal signal collected. We made the same conclusions previously.

The results obtained after post-treatment are shown in Figure 18 and Figure 19. Figure 19 shows the obtained module pictures of the photothermal signal. They correspond respectively to 16 mHz, 24 mHz, and 32 mHz frequencies. It shows a better detection of defects, but also a sensitivity to the patterns of the pictorial layer.

Figure 19 shows the obtained phase images at the same frequencies. On the one hand, they show the possibility of detecting all the defects located in the sample studied. On the other hand, they show a significant rejection of the optical effects induced by the pictorial layer, which was the desired objective. This confirms on the first part the previous theoretical and experimental results. This demonstrates the high interest of a PPT-type analysis in assisting in the conservation of wall paintings.

## 5. Conclusions

In this work, we approached the contribution of a Pulse Phase Thermography type to improve the application during the restoration and conservation process of works of art, and more specifically in mural painting applications. 

First, we underlined that stimulated infrared thermography was already very effective for assistance in the restoration and conservation, allowing the detection of delamination or chemical contamination located in murals paintings (salts for example). The difficulty is that the effect of the pictorial layer does not absorb the excitatory flux in homogeneous ways. This difficulty induces the detection of artifacts signal that conduct to a false identification of defects. This ghost defect should be removed in order to ensure the delivery of the right state of conservation.

To solve this problem, we have suggested studying the contribution of a Pulse Phase Thermography (PPT) analysis.

Firstly, a simulation approach of the photothermal experiment has been developed, using finite element methods to solve the heat equation. We then implemented an experimental study of two representative samples of heritage mural paintings to demonstrate the efficiency of the approach. 

In all three cases, we studied the raw thermograms and then the amplitude and phase maps given by a Fourier Transform. These cases, the raw thermograms and the amplitude maps, allow a good detection of localized failures in the samples studied. However, they remain very sensitive to the different surface radiative properties. 

On the other terms, phase maps show, both theoretically and experimentally, a good detection of defects, but also a very good rejection of these optical problems. 

These results are very encouraging. They seem to improve the non-destructive testing of cultural heritage using stimulated infrared thermography. Further studies in order to generalize these results and to confirm them must be carried by in situ analyses.

## Figures and Tables

**Figure 1 jimaging-05-00072-f001:**
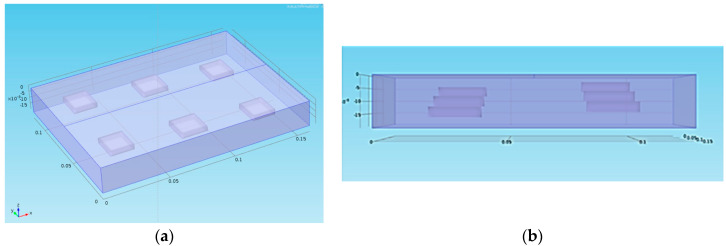
The studied sample: Overview (**a**) and view from the side (**b**) of the theoretically studied sample.

**Figure 2 jimaging-05-00072-f002:**
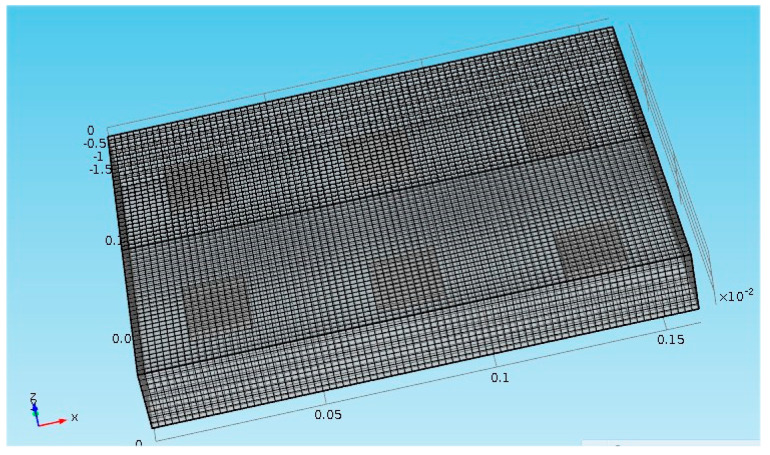
Meshing of the sample studied.

**Figure 3 jimaging-05-00072-f003:**
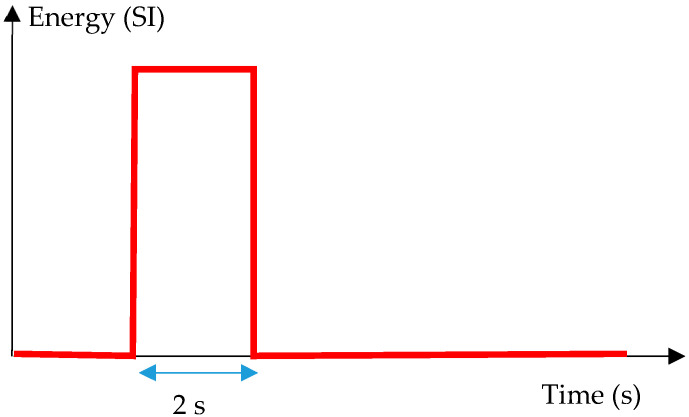
Excitation signal.

**Figure 4 jimaging-05-00072-f004:**
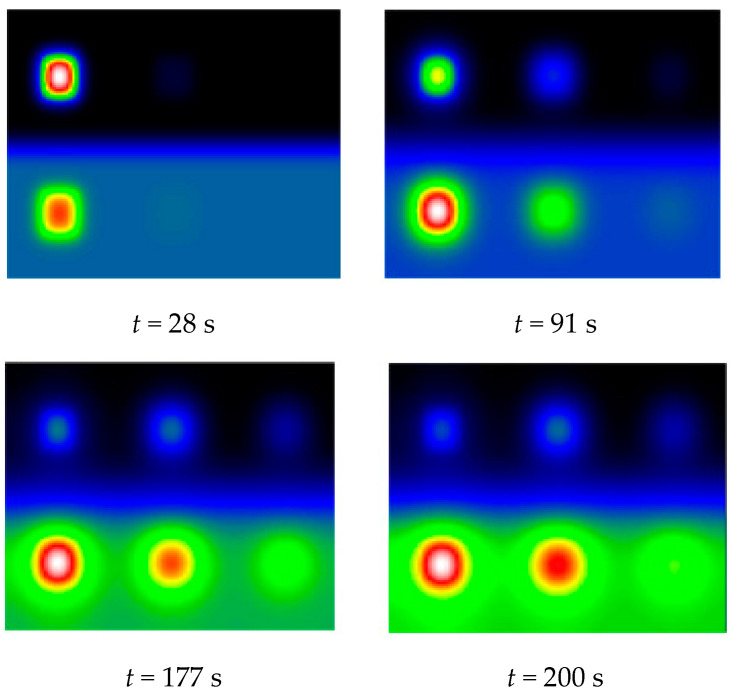
Example of theoretical thermograms obtained (t = 28 s, t = 91 s, t = 177 s and t = 200 s).

**Figure 5 jimaging-05-00072-f005:**
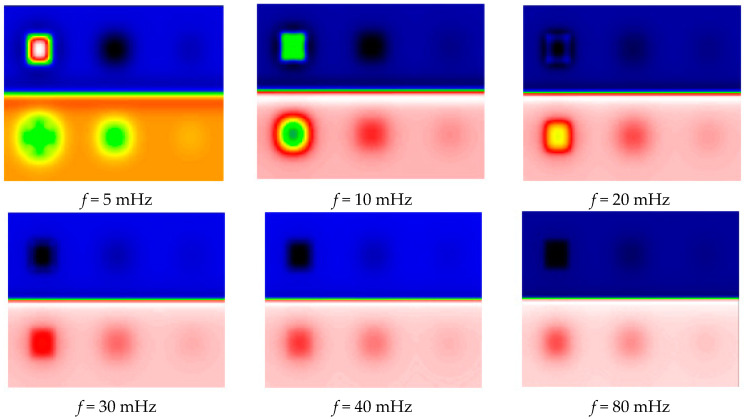
Amplitude images of the calculated harmonic response.

**Figure 6 jimaging-05-00072-f006:**
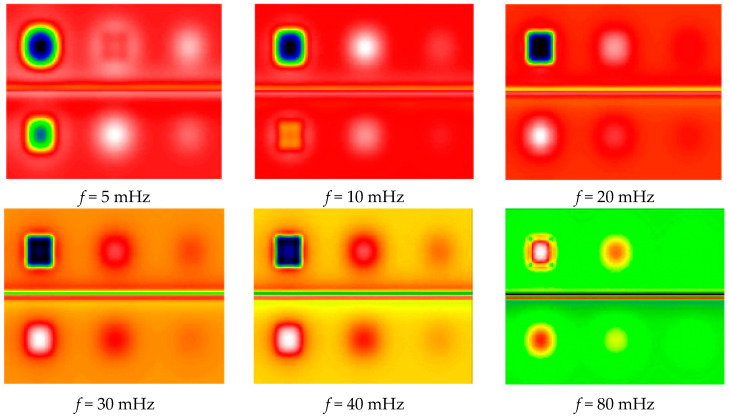
Phase images of the calculated harmonic response.

**Figure 7 jimaging-05-00072-f007:**
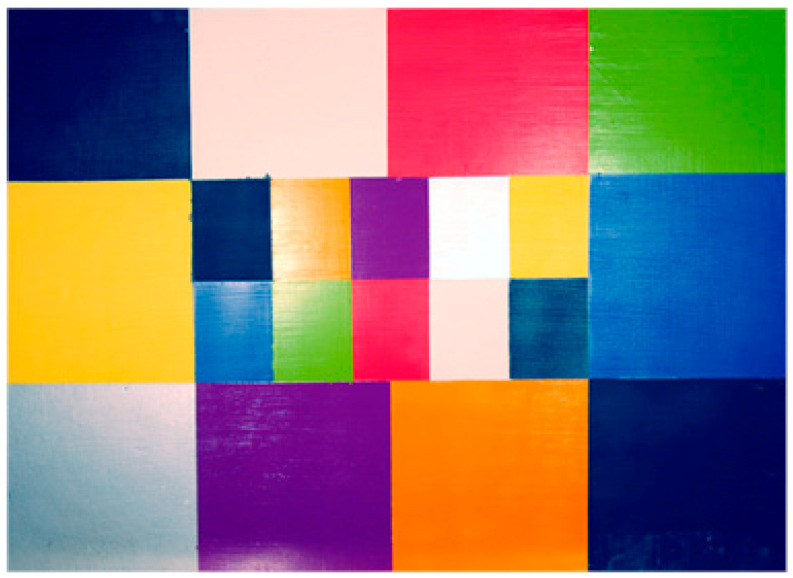
The sample analyzed: Front panel.

**Figure 8 jimaging-05-00072-f008:**
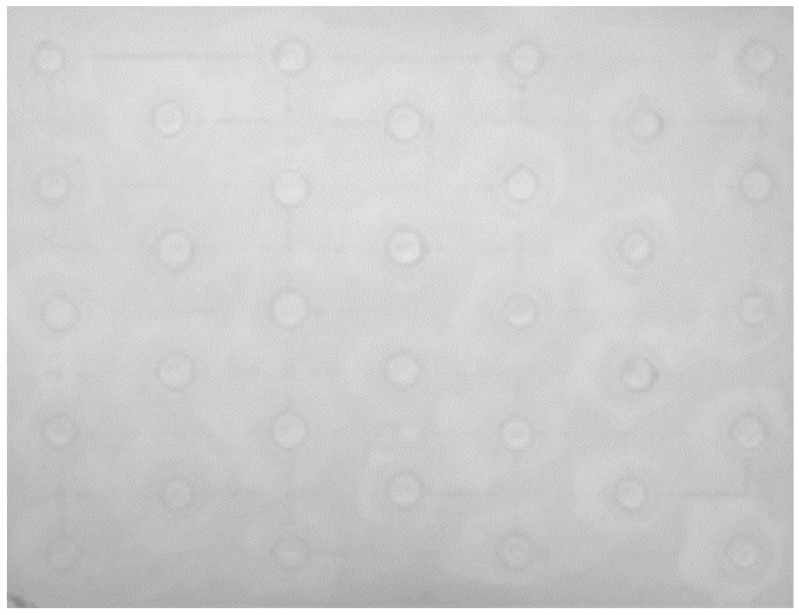
The sample analyzed: Rear side.

**Figure 9 jimaging-05-00072-f009:**
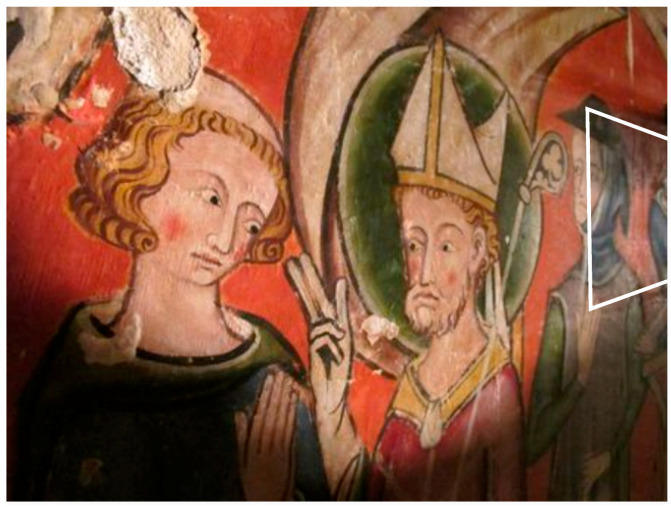
Mural painting “Saint Maurille en Evèque”.

**Figure 10 jimaging-05-00072-f010:**
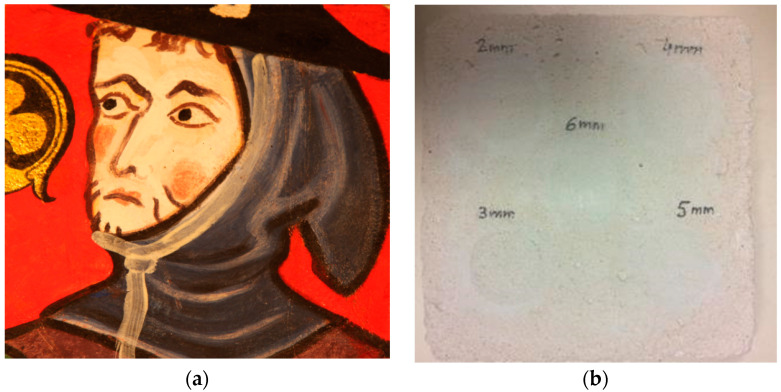
The copy of the mural painting “Saint Maurille en Evèque” studied: Front panel (**a**), back side (**b**).

**Figure 11 jimaging-05-00072-f011:**
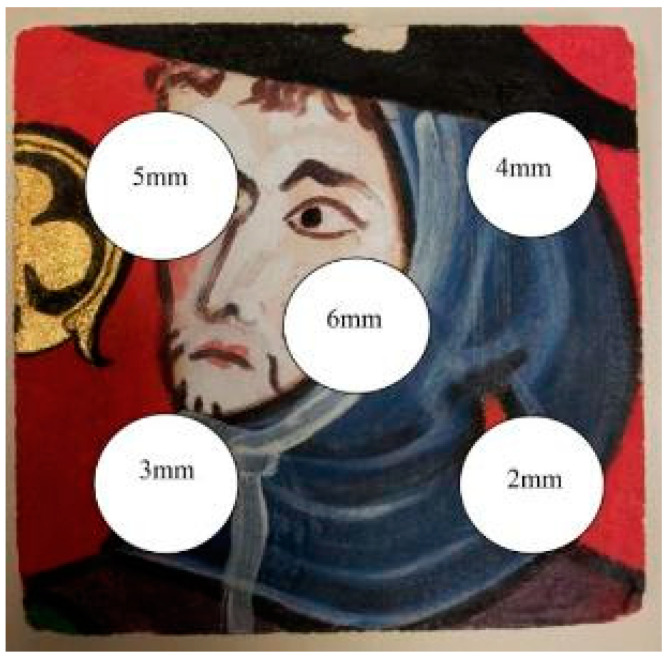
The copy of the mural painting “Saint Maurille en Evèque” studied: Positioning of defects.

**Figure 12 jimaging-05-00072-f012:**
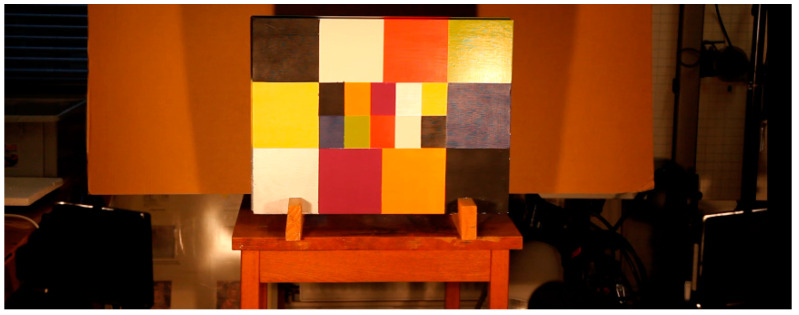
The sample under analysis.

**Figure 13 jimaging-05-00072-f013:**
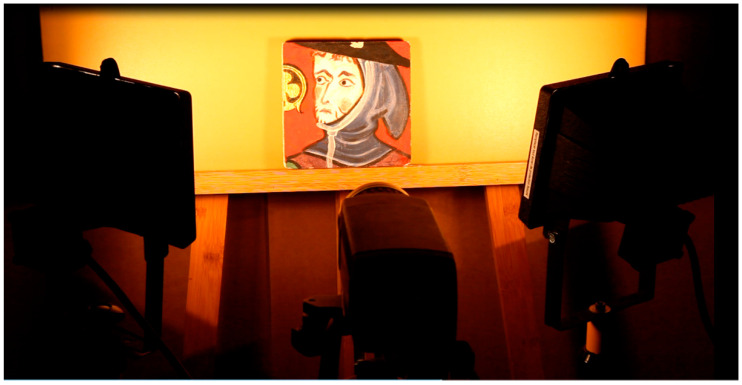
The copy of the mural painting “Saint Maurille en Evèque” under analysis.

**Figure 14 jimaging-05-00072-f014:**
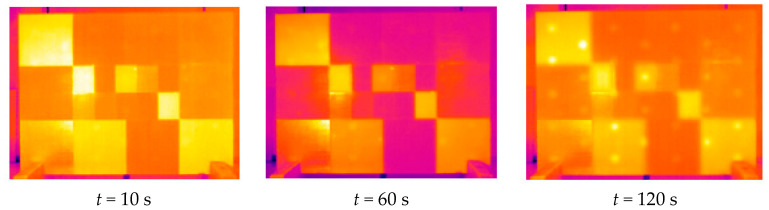
Example of experimental thermograms obtained (at t = 10 s, t = 60 s and t = 120 s).

**Figure 15 jimaging-05-00072-f015:**
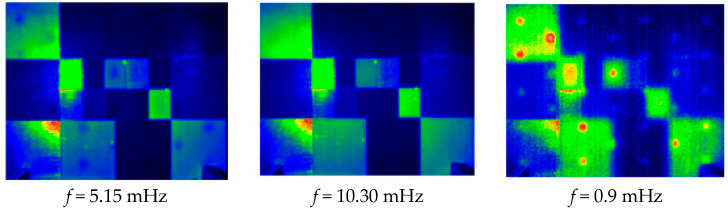
Module images of the calculated harmonic response.

**Figure 16 jimaging-05-00072-f016:**
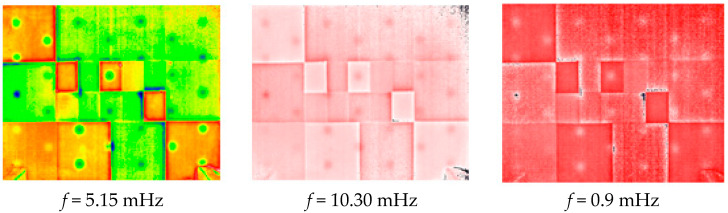
Phase images of the calculated harmonic response.

**Figure 17 jimaging-05-00072-f017:**
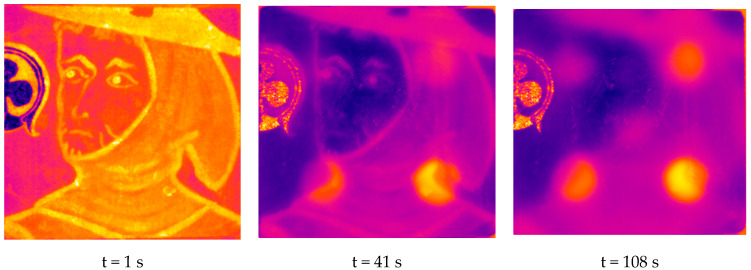
Example of experimental thermograms obtained (at t = 1 s, t = 41 s and t = 108 s).

**Figure 18 jimaging-05-00072-f018:**
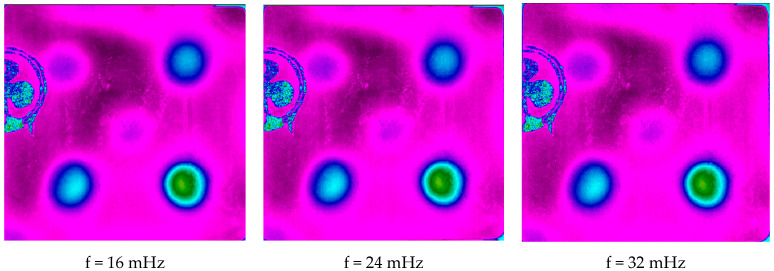
Amplitude images of the calculated harmonic response (at f = 16m Hz, f = 24 mHz and f = 32 mHz).

**Figure 19 jimaging-05-00072-f019:**
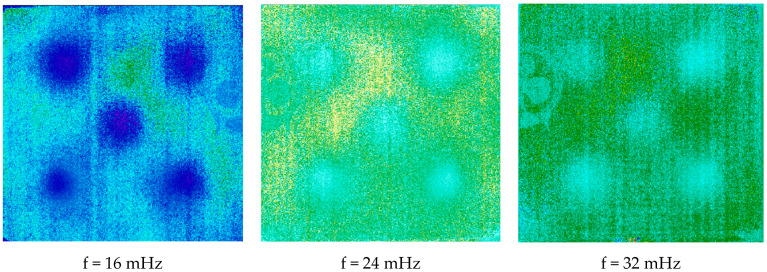
Phase images of the calculated harmonic response (at f = 16 mHz, f = 24 mHz and f = 32 mHz).

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
