# Peer review of "Improvement of the Non-Destructive Testing of Heritage Mural Paintings Using Stimulated Infrared Thermography and Frequency Image Processing"

_2313-433X, 2019, doi:10.3390/jimaging5090072_

Round 1

Reviewer 1 Report

The authors present a study about the use of Pulsed Infrared Thermography to reduce the artifacts introduced by the different adsorption degree of the colors. To prove the validity of their method they perform both, a simulation and a case-study analysis.

The proposed idea is interesting and seems to be supported by the experiments. Nevertheless, the paper has some shortcomings that have to be addressed:

A proper discussion about the literature on the field is totally absent. I expected to read about common methodologies to perform this kind of analysis, by focusing on pro and cons. Moreover, previous studies with PPT based methods could be reported and then the novelty of the proposed strategy highlighted.

There is no explanation on how to read most of the figures. What does it mean the color scale in Figure 3,4,5, 9 etc.? Although in some cases it can be intuitive to understand it, It Is necessary provide a proper description.

 At row 49, the authors introduce Fourier analysis and claim “Fourier analysis … indicates that any mathematical function can be modelled by a sum of trigonometric functions”. This is vague and in general is false. The main tools to perform Fourier analysis are Fourier series and Fourier transform. In equation (1) Fourier series expression is reported. However, Fourier series can be applied only for periodic functions, under the Dirichlet conditions. So not for “any mathematical function”. Moreover, the authors do not explain at all the equation (1). What is n? What are an and bn? What is ω? Although this formula is popular, the equations have to be explained for any readers. Finally, no references are reported.

However, the authors use Fourier transform, so the Fourier series formula is not well-motivated.

Equation 2 is not properly explained.

Please replace the ellipses “…” with “and so on”

 Row 44: “point of vue”-> “point of view“

The English should be revised. I recommend a careful proofreading.

Overall, the work is interesting and could be published on JIMAGING. However, at the current state, a major revision is required. I strongly recommend the authors to address my comments, especially (1), (2) and (3).

Author Response

Hello, about your questions.

Q1) What does it mean the color scale in Figure 3,4,5, 9 etc.?

A1)The color scale represents in one case temperature variations in the case of raw films, or amplitude and phase variations when the PPT has been applied.
In this study we are interested in the relative variations of these quantities, not really their real values. We believe that true values do not provide any interesting information.

Q2) At row 49, the authors introduce Fourier analysis and claim “Fourier analysis … indicates that any mathematical function can be modelled by a sum of trigonometric functions”. This is vague and in general is false....

A1) Indeed it is true, we are talking here about Fourier transform and more precisely about discrete Fourier transform. This is a mistake and we have rectified this point by explaining some theoretical aspects and inviting the reader to refer to the original article on the PPT :

          Maldague, X., & Marinetti, S. (1996). Pulse phase infrared thermography. Journal of applied physics, 79(5), 2694-2698.

Q3) Equation 2 is not properly explained.
A3)This equation describes the passage between the impulse response and the harmonic response of a signal via a Fourier transform. In order not to overload the manuscript we preferred to remove this theoretical expression.

Q4)The English should be revised. I recommend a careful proofreading.

A4)The text has been completely revised. We have corrected the parts that were problematic.

Reviewer 2 Report

In the Reviewer opinion the research paper entitled “Improvement of the non-destructive testing of heritage wall paintings by infrared thermography stimulated by frequency image processing” is very interesting and good.

In this work authorsapproached the contribution of a PPT-type frequency analysis to flash infrared thermography to improve assistance in the restoration and conservation of works of art.

Some comments which greatly enhance the understanding of the paper and its value are presented below. Specific issues that require further consideration are:

The title of the manuscript is well matched to its content.

The structure of the manuscript is proper.

The Introduction sufficiently covers the cases.

In the Reviewer’s opinion, the current state of knowledge relating to the manuscript topic has been covered, but not clearly presented.

An analysis of the manuscript content and the References shows that the manuscript under review constitutes a summary of the Author(s) achievements in the field.

In the Reviewer’s opinion, the bibliography, comprising 11 references, is not representative and exhaustive.

Extensive editing of English language and style required .

I suggest expanding the conclusions.

In the Reviewer’s opinion the manuscript should be published in the journal after corrections.

Author Response

Thank you for reviewing, I'm going to answer to your suggestions.

Q1) In the Reviewer’s opinion, the bibliography, comprising 11 references, is not representative and exhaustive.
A1)The bibliography was missing, it was completed.

Q2)Extensive editing of English language and style required. I suggest expanding the conclusions.

A2) The text has been reviewed and English proofread.

Reviewer 3 Report

Manuscript # 505771 with title “Improvement of the non-destructive testing of heritage wall paintings by infrared thermography stimulated by frequency image processing” aims to investigate the role of the different surface colour layers (and consequently their absorptivity) to the interpretation of IR thermography analysis of wall paintings with the aim to detect hidden defects. Particularly discusses how phase analysis of the acquired data can enhance the recorded information and conclusions.

This is a case application that can enhance the quality of the acquired information and in this respect it is recommended to further discuss the conclusions obtained from the scientific point of view. The authors give specific numbers for the properties of one of the samples (lines 68-69 and 72-73) but these are not discussed anywhere in the text so to get scientific explanations and results. What is the reason for presenting the values of the material properties if no discussion is attempted?

Apart from the scientific level of this paper which definitely needs to be upgraded, the manuscript necessitates significant revision in English. In general it is recommended to avoid the use of “we” “our” etc. and describe the work in passive voice… for example in line 39 instead of “we proceeded to a series of theoretical simulations” it is recommended to revise into “a series of theoretical simulations were employed” and similarly through the manuscript…

Specific points that necessitate revision:

Abstract:

Lines 11-16: the abstract necessitates major revision as it should start with the aim of this work rather than to describe the collaboration between the 3 Institutions and their previous work. The abstract must clearly define the aim of this work and its novelty from the beginning.

Lines 17-18: the 2 sentences need revision as they have issues as regards their grammar and structure :

-“One of these method’s difficulties is its dependence on the painting layers: colours do not absorb the excitatory flux in the same way.” and

“This difficulty can sometimes induce detection artifacts.” (as “artifacts” possibly you refer, correctly, to something observed in a scientific investigation or experiment that is not naturally present but occurs as a result of the preparative or investigative procedure? Please discuss further on this to avoid misunderstandings with the similar term “afterfacts” that means CH objects)

1. Introduction

Lines 35-37. The phrase needs major revision in English. What do you mean with the phrase “energetic nature of the optical effects” and why these induce “disruptive effects”??

Line 44. Revise “briefly lighted” into “briefly illuminated”

Line 48: instead of “from which” use “on which”

Lines 53-54: please rephrase in English the terms “retardation” , also move the “pixel by pixel” before the term “representation”: … “a pixel by pixel representation of…”

Lines 67-68: the article “a” should be avoided, similarly to Lines 72-73

Lines 75-76: Is there a reason that the depths vary between the 2 zones? Please explain further this choice.

Line 78: what is the spectral distribution of your measurements?

Line 87: why the specific values for t, t=28, 177 and 200 s, were chosen?

Line 89: the inhomogeneity was due to the different absorption or the different energy deposition? Please explain further

Line 97: why the specific values of frequencies were chosen? Please discuss further

Lines 105-107: needs revision in English. Not clear i.e. what are the “normal parts”? you mean “reference areas”?

Line 108: ….. “to allow a satisfactory rejection” needs revision in English.

Figure 6: needs revision as the colours are not clearly visible. i.e. they are all more yellow than it should be and silver paint is not obvious. A reference colour palette during the capturing of the photo would enhance the final image. Also although the dimensions are mentioned before a scale would enhance the figure.

Figure 8: what is the new info of Fig 8 in comparison to Fig 6? Possibly the two images can be merged into one?

Line 164: what is a “significant rejection”? please explain further and revise in English

Line 172: "Cycle pein”t, is this correct?

Line 175: what do you mean by the phrase “consistent examples oi oil-painted murals”?

Line 187: Here the name of the restorer should not be mentioned. In the paper you should discuss that a copy of a detail of the mural was prepared and then in the acknowledgements you should mention the name of the person that made the copy…

Line 193: the geometric dimensions of this sample are 15 cm both in length and width..

Lines 227-240: The observed results could be further explained in a more scientific way rather than a brief report.

5. Conclusion the whole section must be significantly revised in English content and syntax.  

Author Response

Thank you for reviewing this manuscript, which has some weaknesses, I agree.

"The authors give specific numbers for the properties of one of the samples (lines 68-69 and 72-73) but these are not discussed anywhere in the text so to get scientific explanations and results. "

==> This information is given as an indication, if it had not been given, we could also be reproached for not having given it. 

"Lines 75-76: Is there a reason that the depths vary between the 2 zones? Please explain further this choice."
==> This simulation file was originally prepared for another analysis. The defects on the 2 zones are not at the same depths but these differences do not affect our analysis.

"Line 78: what is the spectral distribution of your measurements?" 
"Line 89: the inhomogeneity was due to the different absorption or the different energy deposition? Please explain further"==> We have not defined the spectral nature in this simulation. We simply applied a power of 1000 W to the lower absorbance zone and 1500 W to the higher absorbance zone. 

 Lines 105-107: needs revision in English. Not clear i.e. what are the “normal parts”? you mean “reference areas”? ==> corrected. This is the background without defects (reference areas). 

"Figure 6: needs revision as the colours are not clearly visible. i.e. they are all more yellow than it should be and silver paint is not obvious." ==> The white balance of the image has been corrected.

"Line 164: what is a “significant rejection”? please explain further and revise in English" ==> We observe that the optical effects of the paint layer are attenuated.

"Line 175: what do you mean by the phrase “consistent examples oi oil-painted murals”?" ==> rephrased

"Line 187: Here the name of the restorer should not be mentioned." ==> Removed.

"Line 193: the geometric dimensions of this sample are 15 cm both in length and width.." ==>

width and length of 15 cm

"Conclusion the whole section must be significantly revised in English content and syntax. "==> I am aware of the imperfections in terms of English, an effort has been made in this direction.

Round 2

Reviewer 1 Report

The authors have properly addressed my comments and strongly improved the manuscript. In my opinion the paper is overlall on the average and can be accepted for the publication.

Author Response

Hello, I would like to inform you that English has been completely corrected. I received help for this from my co-authors who are more experienced than I am in the field of English writing.
Greetings.

Reviewer 2 Report

In my opinion article should be published in teh Journal.

Author Response

(The authors gave the same response as above.)

Reviewer 3 Report

Reading the responses to the comments of this review the following issues must be stressed:

A)      only selected points have been answered (i.e. the suggested changes on the structure of the abstract have not been performed).

B)       It is recognized that the main author is not a native English speaker but nevertheless the English syntax and context remains poor.

C)       The responses luck the flavour and spirit of a letter that follows the Academic principles. For example the answers

a. --This information is given as an indication, if it had not been given, we could also be reproached for not having given it…” Or

b. ---This simulation file was originally prepared for another analysis. The defects on the 2 zones are not at the same depths but these differences do not affect our analysis.

do not follow an academic profile and definitely do not correspond to the Scientific profile of some of the Authors.

Overall, I wouldn’t recommend its publication prior to a good revision that would revise significantly the English context of the manuscript and would approach in a scientific way the phenomena associated with this imaging methodology and the specific diagnostic challenge so to justify their applicability.

Author Response

(The authors gave the same response as above.)

Round 3

Reviewer 3 Report

Dear Editor, dear Authors,

Reading the revised version of manuscript 505771 it is indeed obvious that a significant effort has been given in order to revise the English content of this paper.

Nevertheless my initial argument has not been responded and no discussion of the observed phenomena and differences in the thermal response of the paints is attempted, which would also justify the presence of the material properties and would give to this paper a qualitative character.

In case this manuscript is accepted here are a number of further revisions:

Abstract Line 11: the term “assistance” is not correct. It is recommended:
Within the framework of conservation and diagnosis for the needs of restoration interventions of cultural property… Abstract Line 15: similarly: ....A diagnostic method has been developed … Line 44: ..can lead to induce the presence of artifacts Line 46: instead o “by the way of” use “using” Line 51: remove the “on it”, “sensitive to heterogeneities of energy deposition on it. Line 60: the obtained results are presented Line 95-96: please revise… Lines 205-206 please revise Line 284. Following this first encouraging result we then focused… Line 325: a simulation approach ….has been developed